# LEARNING UI-TO-CODE REVERSE GENERATOR USING VISUAL CRITIC WITHOUT RENDERING

## ABSTRACT

Automated reverse engineering of HTML/CSS code from UI screenshots is an important yet challenging problem with broad applications in website development and design. In this paper, we propose a novel *vision-code transformer* (ViCT) composed of a vision encoder processing the screenshots and a language decoder to generate the code. They are initialized by pre-trained models such as ViT/DiT and GPT-2/LLaMA but aligning the two modalities requires end-to-end finetuning, which aims to minimize the visual discrepancy between the code-rendered webpage and the original screenshot. However, the rendering is non-differentiable and causes costly overhead. We address this problem by actor-critic fine-tuning where a *visual critic without rendering* (ViCR) is developed to predict visual discrepancy given the original and generated code. To train and evaluate our models, we created two synthetic datasets of varying complexity, with over 75,000 unique (code, screenshot) pairs. We evaluate the UI-to-Code performance using a combination of automated metrics such as MSE, BLEU, IoU, and a novel htmlBLEU score. ViCT outperforms a strong baseline model DiT-GPT2, improving IoU from 0.64 to 0.79 and lowering MSE from 12.25 to 9.02. With much lower computational cost, it can achieve comparable performance as when using a larger decoder such as LLaMA.

## 1 INTRODUCTION

Recent Language Models (LMs) have demonstrated a remarkable capability in generating coherent code. For example, Codex Chen et al. (2021), GPT-4, Code Llama Rozière et al. (2023), and CodeRL Le et al. (2022) have shown promise in aiding software engineers in their daily work. On the other hand, attention-based models have achieved success in various vision tasks, such as image classification, segmentation, and annotation. Among them, some landmark works are Vision Transformer (ViT) Dosovitskiy et al. (2020), Swin Liu et al. (2021), and other architectures, with a few specifically targeting document-related tasks, e.g., Document Image Transformer (DiT) Li et al. (2022a).

In this paper, we take the first step towards reverse-engineering a UI screenshot, i.e., generating an HTML/CSS code that can reproduce the image. By combining the strengths of both LLMs in code generation and Vision Transformer in image representation, and aligning them for the above task, we investigate the possibility of generating the markup code from the visual representations of the original image. Our contributions are threefold. *First, we develop a synthetic dataset generation module used for front-end UI screenshot images and corresponding code generation.* The module is designed to generate images with varying complexity and styles, as well as the corresponding markup code. This dataset is used for training and evaluating our proposed approach and baselines. We also propose htmlBLEU, a more accurate HTML and CSS code similarity metric.

*Secondly, we propose a vision-code transformer (ViCT) architecture composed of a vision encoder and a language model decoder.* We then evaluate the performance of several choices for the encoder and decoder for this task. Specifically, we experiment with GPT-2 Radford et al. (2019) and LLaMA Touvron et al. (2023a) as the text decoder for code generation, and compare the performance of ViT and DiT as image encoders. ViT is a widely utilized model trained on natural images for image recognition tasks. In contrast, DiT is specifically designed for document-related tasks, whose

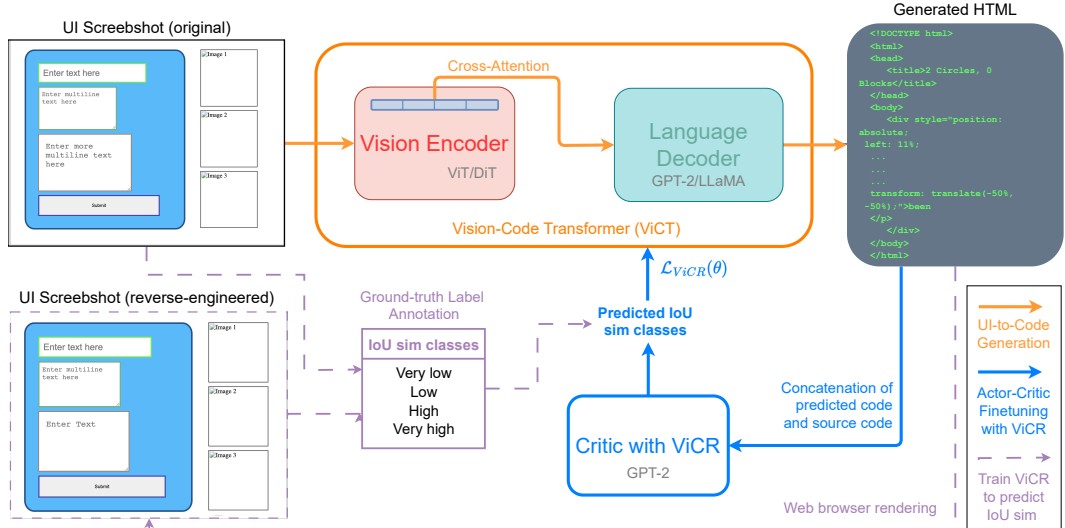

Figure 1: Schematic representation of the proposed *vision-code Transformer* (**ViCT**). ViCT applies a vision encoder-language decoder model to a webpage screenshot and generates the code that aims to reproduce the input screenshot. We finetune ViCT by minimizing the visual discrepancy between the original and reverse-engineered UI screenshots. To this end, we train a *visual critic without rendering* (**ViCR**) as a classifier to measure the visual discrepancy given the generated code as the only input. The training labels for ViCR are generated by rendering the UI image via a browser and computed as the Intersection over Union (IoU) bracket between the original screenshot and the rendered image. Actor-critic fine-tuning is then applied to minimize loss $\mathcal{L}_{ViCR}(\theta)$, with ViCT as the actor with parameter $\theta$ and ViCR as the critic.

domain tends to be closer to that of our task. We explore the efficacy of these architectures in generating HTML/CSS code from webpage screenshots.

*Thirdly, we develop a novel visual critic without rendering (ViCR) used to finetune ViCT for step-by-step code generation.* In particular, we train a critic model that aims to evaluate the discrepancy between the original UI screenshot and the one to be created by the generated code without rendering from the code. ViCR avoids the non-differential rendering loss and its induced overheads. We then apply an Actor-Critic algorithm (AC2) to train ViCT in an end-to-end manner.

Our work, for the first time, establishes a robust baseline for generating markup code from images using vision-code trans-formers. Moreover, we developed a novel evaluation metric htmlBLEU to assess the task. Our proposed approach holds potential applications in front-end web development, as it could offer a more efficient and automated method for generating markup code for web designers.

Table 1: RUID and RUID-Large datasets created in this paper. RUID combines basic HTML elements, while RUID-Large incorporates all HTML elements, providing more complex and similar examples to real user interfaces.

|  | RUID | RUID-Large |
|---|---|---|
| **Samples** | 25000 | 50000 |
| **Colors** | Arbitrary | Arbitrary |
| **Objects** | 6 | 12 |
| **Words** | 1 | 5 |
| **Elements** | Rectangle, Eclipse, Button, div | a, img, div, span, button, text, textarea, input, submit, radio, select, p, checkbox |

## 2 RELATED WORKS

Recent years have witnessed significant progress in both image-understanding and text-generation tasks, empowered by deep learning and the availability of massive datasets Goodfellow et al. (2014); Radford et al. (2016); Esser et al. (2021); Sutskever et al. (2014); Graves et al. (2013);

Brown et al. (2020). Transformer models trained on large-scale data in a self-supervised manner have played a key role in these advancement Vaswani et al. (2017); Devlin et al. (2019).

Code prediction and generation have received growing attention in the realm of text generation. While traditional NLP methods like N-grams and Probabilistic Context-Free Grammar (PCFG) have encountered challenges in code generation tasks Maddison & Tarlow (2014), recent advancements in Transformer models have led to remarkable improvements. For instance, Codex Chen et al. (2021) and its derivative tool Copilot, fine-tuned on publicly available code from GitHub, have achieved impressive performance on code generation tasks.

InCoder Fried et al. (2022) enables bidirectional context for code infilling by training on publicly available repositories where code regions have been randomly masked and moved to the end of each file. CodeGen Nijkamp et al. (2022) explores a multi-step paradigm for program synthesis, dividing a single program into multiple subproblems specified by multiple prompts. CodeRL Le et al. (2022) incorporates deep Visual Critic with an error-predictor critic network that generates rewards by classifying the code.

Works like BLiP Li et al. (2022b), Git Wang et al. (2022), and CoCA Yu et al. (2022) have leveraged large-scale pre-training on visual-textual data followed by fine-tuning on target tasks and outperformed traditional methods. Recent works such as BLiP 2 Li et al. (2023), and derivatives such as Minigpt-4 Zhu et al. (2023) and InstructBLIP Dai et al. (2023) significantly improve the text generation capacity by employing larger language models such as Vicuna Chiang et al. (2023) and LLaMA-2 Touvron et al. (2023b).

Other models aiming to generate code from images include pix2code Beltramelli (2018), which generates code based on context and GUI images, and Sketch2code Robinson (2019), which attempts to generate code from wireframes using traditional computer vision and deep learning algorithm. The paper found the deep learning-based pipeline to perform better. While some works add components such as image style transfer, they rely on predefined classification for code generation.

Another work Pix2Struct Lee et al. (2022) uses a novel screenshot parsing objective to generate a simplified HTML parse from a masked screenshot of a webpage, effectively learning rich representations of the underlying structure of web pages. It encourages joint reasoning about the co-occurrence of text, images, and layouts in webpages.

In contrast, our work builds upon these foundations and offers innovative approaches to generating HTML/CSS code. We employ transformer architecture and introduce a visual similarity signal in the training process, enhancing the accuracy and versatility of our model. Furthermore, we curate a diverse dataset tailored for this specific task and establish a strong baseline for generating markup code from images using vision-code transformers, contributing to the solving of this challenge. Additionally, we introduce a novel evaluation metric designed to facilitate a more accurate assessment of the task's performance.

## 3 METHODOLOGY

In this section, we elucidate the components of our approach, including the dataset, evaluation metrics, baseline training procedure, task formulation using actor-critic, and details of our experiments.

### 3.1 NEW DATASETS FOR UI TO CODE GENERATION

While a variety of code generation datasets are available Rozière et al. (2023) the space of open UI-code correspondence datasets is scarce. Due to this, we utilize a synthetic generation process to create two datasets of varying complexity. We generate Random UI Dataset (RUID) by combining a small number of HTML elements, such as two types of Divs, a square, a circle, and a button element, with randomly chosen style attributes. This results in a diverse set of images that can be used for the training process. We then create RUID-Large, which incorporates elements in RUID but expands it to most tags available in HTML, randomly generating trees with forms, divs, inputs, dropdowns, etc.

All the synthetic dataset code is enclosed in standard HTML opening and closing tags, specifically:

```
<!DOCTYPE html>
```

```
2 <html>
3   <head>
4     <title>{title}</title>
5   </head>
6   <body>
7     {elements}
8   </body>
9 </html>
```

For RUID We set the description of the elements present in the body as the title, for example, "2 Circles, 0 Blocks". For each element, a paragraph containing a number of words has been added, with the text sourced from Project Gutenberg Project Gutenberg (2023). The dataset generator can be used to create samples of varying complexity. We focus on sampling small elements for the RUID dataset. Most of our experiments are performed on the RUID dataset.

A summary of the settings used for the work can be found in Table 1. Overall, the maximal input length of the adopted models acts as a ceiling to the length of the generated code.

An example of the generated element is below. Note that some of the parameters in Table 1 have been adjusted to fit the webpage, but the aesthetics of the generated elements have not been taken into account.

The datasets are used with a split of 80:10:10 for training, validation, and testing. For each code sample, we take a screenshot of its generated webpage as it looks when opened in a Chromium browser. Thereby, we collected a dataset of (image, code) pairs, facilitating the training and evaluation of our models.

It is worth noting that the traditional pipelines cannot directly address the task studied in this paper due to their different problem formulations and dataset formats. Specifically, they reduce the problem to classification between predefined building blocks while our approach focuses on free-form code generation. Hence, comparisons to them on our proposed dataset and task are infeasible. That being said, we create baselines for Transformer models on our dataset for HTML/CSS code generation that explores greater freedom in attributes and has no restrictions on color variety. In addition, we have evaluated the performance of general Visual Language Models such as InstructBlip Dai et al. (2023), minigpt-4 Zhu et al. (2023), and Bing chat in executing the specified task.

Table 2: Element types, widths, and parameters used for the synthetic dataset generation. The number of elements per sample was randomly drawn from 1 to 6.

| Properties | Rectangle | Ellipse | Button |
|---|---|---|---|
| Left (%) | 0-80 | 0-80 | 0-80 |
| Top (%) | 0-80 | 0-80 | 0-80 |
| Width (%) | 10-30 | 10-30 | 10-30 |
| Height (%) | 10-30 | 10-30 | 10-30 |
| Background | Uniform | Uniform | – |
| Text Length | 1 Word | 1 Word | 1 Word |
| Occurrence | 12/25 | 12/25 | 1/25 |

### 3.2 PROPOSED VISION-CODE TRANSFORMER (VICT)

Our model employs a Visual Transformer (ViT) Dosovitskiy et al. (2020) as the vision encoder. The ViT processes the input image by dividing it into patches and encoding them as a sequence of tokens, supplemented by a [CLS] token representing the image's global features. This computationally efficient approach, which has become widely adopted in recent methods such as Li et al. (2021), eliminates the need for pre-trained object detectors for visual feature extraction. Since ViT was usually pre-trained on natural images while the images in our dataset are mainly UI images, we further explored the DiT model Li et al. (2022a), a transformer trained on document images (which has a smaller domain gap to UI images), as an alternative image encoder. We use GPT-2 Radford et al. (2019) and LLaMA Touvron et al. (2023a), autoregressive language models, as the image-grounded text decoder. This model integrates the ViT encoder's output tokens into its first-layer inputs via a cross-attention (CA) layer, positioned between the causal self-attention (CSA) layer and the feedforward network (FFN) of the text encoder. The input sequence length is set to 900 while each sequence begins with a [BOS] token and is terminated by a [EOS] token. By optimizing a cross-entropy loss function, we train the ViT encoder and GPT-2 decoder in an end-to-end manner, which maximizes the likelihood of the ground-truth code in an autoregressive way. This objective

provides the model with the ability to generalize and effectively convert visual information into coherent HTML/CSS code.

### 3.3 FINE-TUNING USING VISUAL CRITIC WITHOUT RENDERING (VICR)

Our goal in the finetuning process is to improve the visual similarity between the original and the predicted code samples when rendered. To do this we formulate the image-to-code generation as an RL problem, where visual similarity score, like IoU serves as the basis for the reward signal, while the finetuned encoder-decoder model serves as the stochastic policy, with token predictions as action steps.

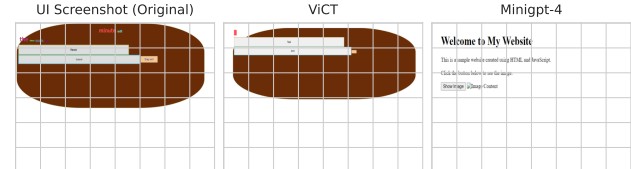

Figure 2: Comparison of ViCT (ours) and miniGPT-4 reverse-engineered images with the original UI screenshot.

$$L_{\text{ViCR}}(\theta) = -\mathbb{E}_{W^s \sim p_\theta}[\text{IoU}(I_{\text{org}}, I_{W^s})] \tag{1}$$

where $\theta$ are the parameters of our model. $\mathbb{E}_{W^s \sim p_\theta}$ represents the expectation over synthetic samples $W^s$ drawn from the policy $p_\theta$, which is the distribution over actions defined by the model. $\text{IoU}(I_{org}, I_{W^s})$ is the Intersection over Union (IoU) score, a measure of the visual similarity between the input image $I_{org}$ and the image $I_{W^s}$ rendered from the synthetic sample $W^s$.

Due to challenges in the training in text generation setups Zhong et al. (2017) Le et al. (2022) we modify the actor-critic apporach used by CodeRL Le et al. (2022) where a language model is used as the critic. We experiment with using GPT2 and BERT Devlin et al. (2019) models. Though from initial results, we see that GPT2 Critic significantly outperforms BERT, with the BERT model collapsing to single class prediction even after oversampling. So we proceed with using GPT2 Critic for the experiments. The performance of the critic can be seen in Figure 4.

The critic model is trained using source code samples from the training set paired with the sampled result from the baseline model as input, and the similarity score between respective rendered visualizations as the label. The inputs are concatenated using the following template: f"{predicted_code}\nGround:{source_code}".

To simplify the training process, we do not use raw similarity values, rather approaching the critic training as a classification problem between 4 classes. Specifically, the IoU thresholds used are: very low (0-0.23), low (0.23-0.42), high (42-77), and very high (77+).

We then use the critic model to generate intermediate outputs for each prediction, and source code samples in the training set. We also create a mask to only use the values related to predicted code tokens in the tuning loss calculation. We then apply softmax to each token's corresponding output and select values of the IoU ground truth bucket. The resulting vector is then multiplied by the corresponding assigned reward. We assign rewards of -1, -0.7, and -0.3 to classes 0, 1, and 2, respectively, and a positive reward of 1 to class 3. The resulting vector is used to scale the original loss during the fine-tuning phase using the update:

$$\nabla_\theta \mathcal{L}_{\text{ViCR}}(\theta) \approx -\mathbb{E}_{W^s \sim p_\theta}\left[r(W^s)\sum \hat{q}_\phi(w_t^s)\nabla_\theta \log p_\theta(w_t^s \mid w_{1:t-1}^s, D)\right] \tag{2}$$

Where $\nabla_\theta \mathcal{L}_{ViCR}(\theta)$ represents the gradient of the RL loss function with respect to the model parameters $\theta$. The term $\hat{q}_\phi(w_t^s)$ represents the critic's estimated value for the token $w_t^s$ at time step $t$. $\nabla_\theta \log p_\theta(w_t^s \mid w_{1:t-1}^s, D)$ is the gradient of the log-probability of token $w_t^s$ at time step $t$, given the history of previous tokens $w_{1:t-1}^s$ and additional data $D$, with respect to the model parameters $\theta$.

### 3.4 IMPROVING METRICS

Evaluating Vision-Code models can be challenging. Some of the common metrics for language generation, such as BLEU scores can be misleading since there are multiple ways to achieve the same visual look during rendering. Similar goes for using a large language model such as GPT-4 for evaluating the output. Rendering itself is costly since it depends on an eternal browser. Inspired by

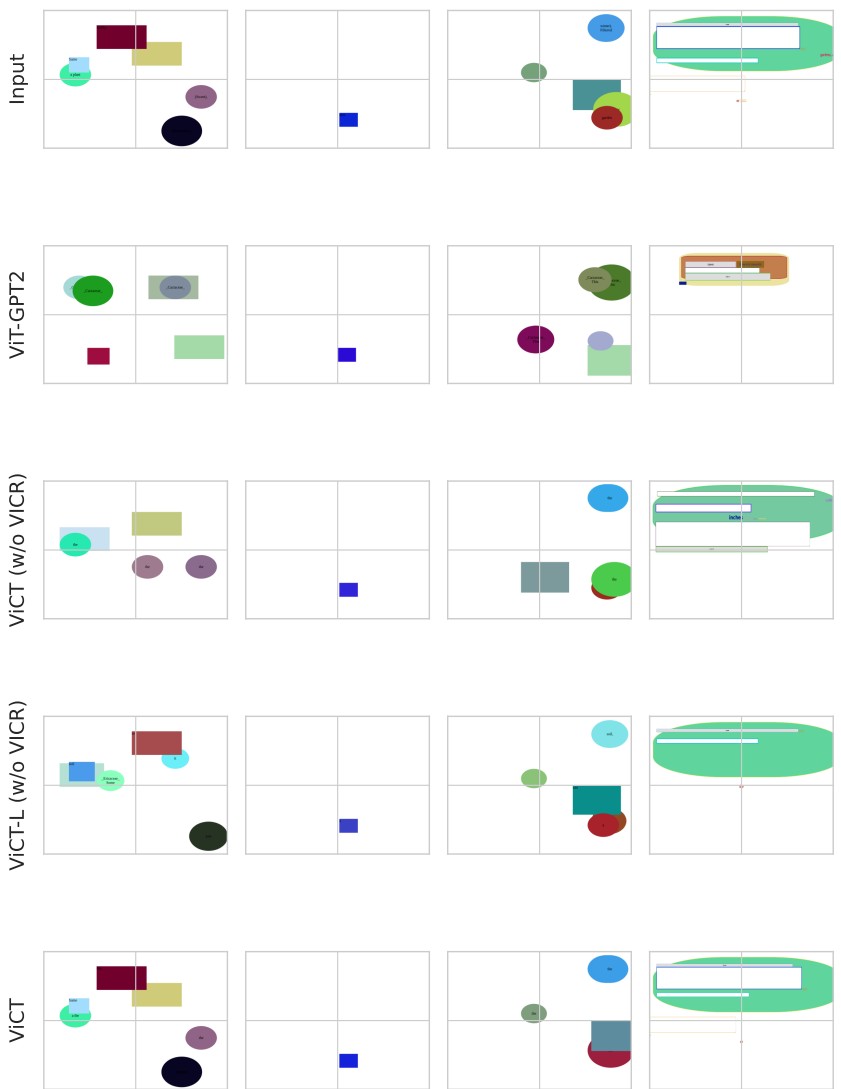

Figure 3: Example renderings from different models tested on the RUID dataset. The top row shows input UI screenshots from the dataset. The subsequent rows show renderings of the code predicted by each model for those specific inputs.

CodeBLEU Ren et al. (2020) to improve generated code evaluation without rendering we propose htmlBLEU metric which emphasizes important pieces of code as well as aligning attributes and elements in the DOM tree.

Still, the metrics are a proxy for target, which is a human assessment of quality. To measure this we conducted a survey with 59 volunteers. The participants were presented with the original screenshot and the rendered one side-by-side and then asked to rate them on a scale of 0 to 100 in terms of structural and color similarity. For each model, we report the averaged min-max normalized score across all annotators and samples.

For automated metrics, we employed two groups to assess the model performance: (1) code-based metrics, which compare the generated code against the original code producing the input screenshot; and (2) image-based metrics, which evaluate the screenshot of the generated images against the input.

We measure image similarity (2) in the following ways. First, we calculate the mean squared error (MSE) between the pixel values of the two images. Second, we create binary masks for each image,

setting pixels to 1 wherever the magnitude is greater than zero. We then compute the MSE and intersection over union (IoU) between these masks. For (1), we employed BLEU score and a dataset-specific metric called Element Counts, where the presence of all elements in the generated code results in a score of 1, and misalignment yields a score of 0.

Since the BLEU score equally penalizes any differences between the two pieces of code, it is not an ideal metric for code evaluation. To avoid penalizing differences that do not lead to visual discrepancies, we develop a new metric, htmlBLEU, from Code-BLEU Ren et al. (2020), as HTML code lacks data flow or syntactic abstract syntax tree (AST). html-BLEU comprises four components: a basic BLEU score, a weighted BLEU score focusing on the most important keywords for HTML code, a Document Object Model (DOM) Tree Matching between the corresponding HTML elements, and an attribute matching that aims to correspond elements and attributes.

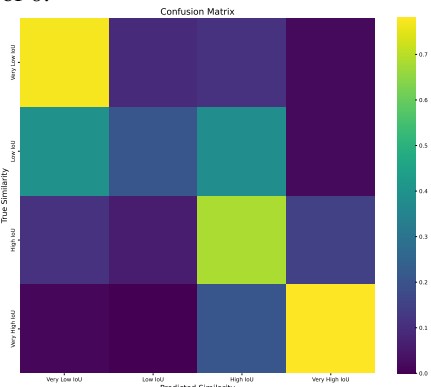

Figure 4: Confusion matrix for ViCR evaluating visual similarity of two code snippets.

To examine the efficacy of htmlBLEU, we measure the Spearman's rank correlation coefficient Spearman (1904) between htmlBLEU scores and the MSE between input and generated images. The correlation between htmlBLEU and MSE is 0.764, compared to 0.329 for the correlation between BLEU and MSE. The significantly higher correlation of htmlBLEU demonstrates that it more accurately reflects visual similarity than BLEU.

## 4 EXPERIMENTS

### 4.1 ESTABLISHING BASELINES

To establish baselines, we tested two recent visual language models - InstructBLIP and Minigpt-4 - on two tasks. The first was identifying the number of distinct shapes in an image. The second was recreating the source code for the image. In our experiments, neither model achieved strong performance on these tasks. They generated unchanging or nonsensical output for the source code and hallucinated the number of shapes. For example, Minigpt-4 produced the same incorrect output regardless of the input image (Figure 2). These results indicate that general visual language models of this type may not be well-suited for source code generation from image inputs without additional training or modifications.

### 4.2 RUID AND RUID-LARGE DATASETS

We report the image similarity and code generation metrics for different models in Table 3, in which the DiT-based model significantly outperforms the ViT-based model across all metrics. Although ViT can accurately handle simpler images with a single element, it struggles to correctly capture > 1 types of elements present in the input image, as well as their positions and colors.

This is consistent with the qualitative results of our study: The ViT-based model was found to frequently miss or misinterpret elements and had difficulty accurately predicting the hexadecimal values of the colors. Figure 3 show some examples: DiT model accurately identifies the types and locations of the elements, while ViT model struggles with these tasks.

The ViCT-L (w/o ViCR) model which uses DiT-Large encoder and GPT2-Large decoder models further improves the performance, demonstrating the benefits of leveraging larger models in the image-to-code generation task. The DiT-Large-GPT2-Large model exhibits enhanced results in terms of IoU, MSE, and element counts compared to the ViCT (w/o ViCR) model.

Moreover, we explore the application of Visual Critic to further enhance the code generation process. Specifically, the ViCR model outperforms its normal finetuning variants in all metrics, showcasing the effectiveness of RL-based approaches in improving the visual similarity between the generated code and the input image. Additionally, when compared to the DiT-Large-GPT2-Large model, the

Table 3: Comparison of various model performances on the RUID dataset, comprising basic shapes and elements. Our model, ViCT exhibits superior performance over ViCT without ViCR finetuning and is competitive with, or exceeds, larger models. The "Metrics" section presents automatically calculated metrics, while "Human Evaluation" provides normalized survey results, both displayed with mean and variance values.

| Model | ViT-GPT2 | ViCT (w/o ViCR) | DiT-GPT2 (L.) | ViCT (Our) |
|---|---|---|---|---|
| **Metrics** | | | | |
| BLEU ↑ | 0.65 ± 0.08 | 0.74 ± 0.09 | 0.68 ± 0.11 | **0.76** ± 0.08 |
| htmlBLEU ↑ | 0.62 ± 0.13 | 0.69 ± 0.14 | 0.67 ± 0.12 | **0.70** ± 0.13 |
| IoU ↑ | 0.31 ± 0.25 | 0.64 ± 0.27 | **0.81** ± 0.19 | 0.79 ± 0.23 |
| MSE ↓ | 19.63 ± 11.59 | 12.25 ± 8.83 | 11.34 ± 8.17 | **9.02** ± 6.96 |
| MSE (Mask) ↓ | 0.15 ± 0.09 | 0.07 ± 0.06 | **0.03** ± 0.05 | **0.03** ± 0.04 |
| Element N ↑ | 0.97 ± 0.16 | 0.97 ± 0.18 | 0.86 ± 0.36 | 0.96 ± 0.20 |
| **Human Evaluation (Normalized)** | | | | |
| Color Fidelity ↑ | 0.41 ± 0.29 | 0.66 ± 0.28 | 0.51 ± 0.27 | **0.83** ± 0.21 |
| Structural Sim. ↑ | 0.49 ± 0.33 | 0.67 ± 0.27 | **0.85** ± 0.18 | 0.83 ± 0.25 |

Table 4: Comparison of various model performances on the RUID-Large dataset, overall scores are lower than RUID due to the dataset being more challenging, incorporating most HTML elements in complex combinations. Still, our model, ViCT performs similar or better than larger models.

| Model | ViT-GPT2 | ViCT (w/o ViCR) | Dit-GPT2 (L.) | DiT-LLaMA | ViCT (Our) |
|---|---|---|---|---|---|
| **Metrics (800 token dataset)** | | | | | |
| BLEU ↑ | 0.60 ± 0.07 | 0.72 ± 0.08 | 0.65 ± 0.10 | 0.72 ± 0.09 | **0.74** ±0.07 |
| htmlBLEU ↑ | 0.59 ± 0.12 | 0.68 ± 0.11 | 0.64 ± 0.13 | 0.66 ± 0.12 | **0.69** ±0.12 |
| IoU ↑ | 0.24 ± 0.20 | 0.38 ± 0.18 | **0.42** ±0.16 | 0.41 ± 0.17 | 0.40 ± 0.19 |
| MSE ↓ | 19.95 ± 10.50 | 15.86 ± 9.20 | 14.90 ± 8.50 | 14.20 ± 8.20 | **13.50** ±7.80 |
| MSE (Mask) ↓ | 0.15 ± 0.08 | 0.11 ± 0.07 | 0.09 ± 0.06 | **0.08** ±0.06 | **0.08** ±0.05 |
| Element N ↑ | 0.92 ± 0.15 | 0.94 ± 0.16 | 0.89 ± 0.18 | 0.91 ± 0.17 | **0.93** ±0.17 |
| **Human Evaluation (Normalized)** | | | | | |
| Color Fidelity ↑ | 0.73 ± 0.10 | 0.77 ± 0.08 | 0.74 ± 0.09 | 0.76 ± 0.09 | **0.81** ±0.07 |
| Structural Sim. ↑ | 0.76 ± 0.12 | 0.81 ± 0.06 | 0.83 ± 0.10 | **0.84** ±0.09 | **0.84** ±0.08 |

ViCR model achieves superior results in certain metrics, indicating the added benefits of RL integration. These findings highlight the potential of RL-based approaches in enhancing code generation and improving the visual fidelity of the generated code.

As shown in Figure 5, both models' performance drops as the code samples get more complex. But DiT suffers a much slower drop on the IoU curve than ViT when predicting more than one element. Another interesting observation is that the generated code does not necessarily have a high text similarity as the ground-truth code for the input screenshot. It can still produce visually similar webpage even if the textual similarity is low, which indicates a promising generalization capability of the models.

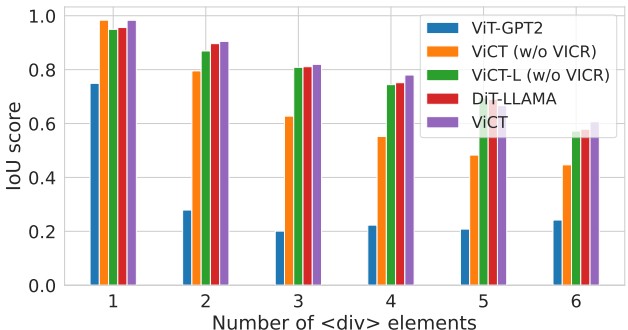

Figure 5: IoU vs. complexity (the number of <div> elements in the ground-truth code). Approximates complexity of reverse generation. Scores for different models tested. IoU drops as more elements are added.

### 4.3 ABLATION

We conducted an ablation study by using cross-entropy (CE) instead of IOU as the metric to create classification labels for the critic model. There is no notable improvements from ViCT (w/o ViCR) scores of IoU of 0.64 or MSE of 12.25 for RUID dataset. In tables 2 and 4 we can see gains of ViCT versus when trained without ViCR. In figure 5 we also see that there is a significant improvement in the drop of performance over increasing sample complexity when ViCR is used.

## 5 CONCLUSION

This paper investigates how to build and train a vision-code transformer for reverse engineering a webpage screenshot and generating the HTML/CSS code that can reproduce the screenshot. We apply ViT or DiT as an image encoder and GPT-2 as a textual decoder that generates code from the ViT/DiT features of the input image. Unlike traditional pipelines, our models can be trained in an end-to-end manner for free-form code generation. Moreover, we collect a synthetic dataset to train and evaluate the proposed models and develop a novel htmlBLEU metric to evaluate the matching between the ground-truth code and the generated one. Our experimental results show that the ViCT (w/o ViCR) model outperforms ViT-GPT2 in terms of multiple metrics and human evaluation. Furthermore, we explore the effect of model size, as well as of Actor-Critic finetuning on the model performance. We train the critic model to consider visual similarity information and modify the actor encoder-decoder network's loss function to incorporate the critic model's output. The results show that the RL finetuning is effective at significantly boosting the underlying model's performance, with the resulting model scoring similarly or better larger sized models on most metrics.

This study serves as a proof-of-concept in the field, demonstrating that Transformer architectures could be a viable end-to-end solution for this task. However, further research is necessary to extend the generated code's length, improve text snippet identification in the image, and explore more complex examples where the corresponding code may not be as straightforward.

## 6 LIMITATIONS

Despite the promising results, it is essential to highlight the limitations of this study. The synthetic dataset used, albeit including variations in the size and location of elements, may not fully encapsulate the complexity of real-world web pages. Moreover, the dataset does not strictly adhere to all front-end development best practices, necessitating further research for practical implementation in real products. Additionally, our current pipeline is solely capable of generating static text pages and is limited to small samples.

Moreover, the Visual Critic pipeline necessitates tuning of certain hyperparameters, specifically learning rate, and rewards, to ensure stable training. Consequently, while our method is efficient, it introduces a degree of overhead when adapting to new datasets and tasks.

## 7 REPRODUCIBILITY STATEMENT

We provide comprehensive details of our experimental setup, with additional information in the Appendix. Our computations were performed on standard CPUs and GPUs using open-source software. An anonymized copy of the research source code is included in a downloadable zip archive. The code is designed to be user-friendly and extensible for future research. Upon publication, we will provide a public link to the source code, model weights, and datasets.

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

## A  APPENDIX

### A.1  BASELINES

In this appendix, we present the output of our experiment using MiniGPT-4 with the Llama 2 7B chat model to generate HTML and CSS code for recreating a visual design based on a given prompt.

| UI Screenshot (original) | ViCT (Our) | Minigpt-4 LLaMA |

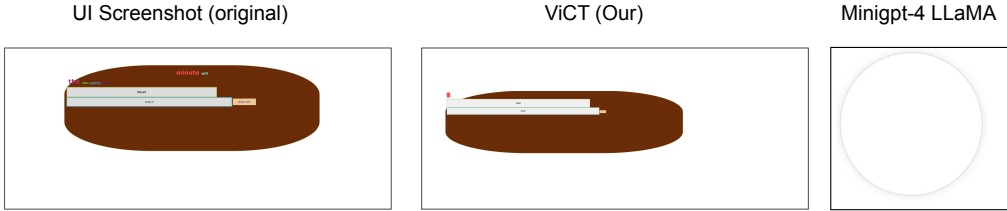

Figure 6: Comparison of our proposed model and Llama 2 chat model backbone on Minigpt-4 for mimicking original website design.

The following prompt was given to the model for code generation:

> Analyze the image provided and extract the visual design elements. Please generate the corresponding HTML and CSS code that recreates the visual design seen in the screenshot. Ensure that the structure and positioning of elements, as well as the color and size of shapes, are accurately represented in the code.

The resulting HTML-CSS code generated by the model has some minor errors. We use GPT-4 to fix the mistakes and beautify the code.

The resulting code is:

```html
<html>
  <div class="container">
    <div class="oval">
      <div class="line-1">
        <div class="line-2">
          <div class="line-3">
            <div class="rectangle">
              <div class="oval-inner">
                <div class="oval-top-corner"></div>
                <div class="oval-bottom-corner"></div>
              </div>
            </div>
          </div>
        </div>
      </div>
    </div>
  </div>
</html>
```

Listing 1: MiniGPT4 Generated HTML Code

```css
.container {
  position: relative;
  width: 200px;
  height: 200px;
```

```
5    border-radius: 50%;
6  }
7
8  .oval {
9    position: absolute;
10   top: 50%;
11   left: 50%;
12   transform: translate(-50%, -50%);
13   border-radius: 50%;
14   box-shadow: 0 0 10px rgba(0, 0, 0, 0.1);
15   background-color: #fff;
16   border: 1px solid #ddd;
17   padding: 20px;
18   width: 150px;
19   height: 150px;
20  }
21
22  .line-1 {
23   position: absolute;
24   top: 50%;
25   left: 0;
26   transform: translateY(-50%);
27   width: 20px;
28   height: 20px;
29  }
```

Listing 2: MiniGPT4 Generated CSS Code

Overall, the minigpt-4 output code seems to capture the general shape (circular) of the input image, however, it struggles with generating any more specific features. We also observe that the model hallucinates a lot with none of the tested randomly selected 10 images correctly approximating the number of distinct shapes in the screenshot.

We also compare the performance of GPT-4 to Bing Chat. Bing Chat, powered by GPT-4 has recently added a feature to understand images. Not much is currently published about the workings of this system. We give it similar samples as before, with Bing chat we get a closer approximation of a number of shapes, over testing 10 samples the model demonstrated 90% accuracy in predicting the correct amount of distinct shapes, a significant improvement over minigpt-4 and Instruct-Blip. However, when reverse-engineering HTML code it struggles to approximate the number, shape, or colors of the original image to a meaningful extent 7.

## A.2 DATASETS

As mentioned in 1 RUID consists primarily of HTML div elements given various shapes and sizes using CSS and buttons, while RUID-Large expands by adding most of the common HTML elements including forms, radio buttons, images, links, etc. In the figure 9, we see some examples from the dataset.

To generate a greater diversity in the samples we allow almost unconstricted generation in terms of number and child object types for any HTML element. Afterwards we use a controller module to detect and remove and illegal generations, such as elements nested within text fields, or or in form input elements.

A Single element from a random sample in RUID dataset:

```
1  <div
2    style="position: absolute; left: 11%; top: 79%; width: 20%;
       height: 20%;
3    background-color: #C6FC54; border-radius: 50%; text-align:
       center;">
4    <p
```

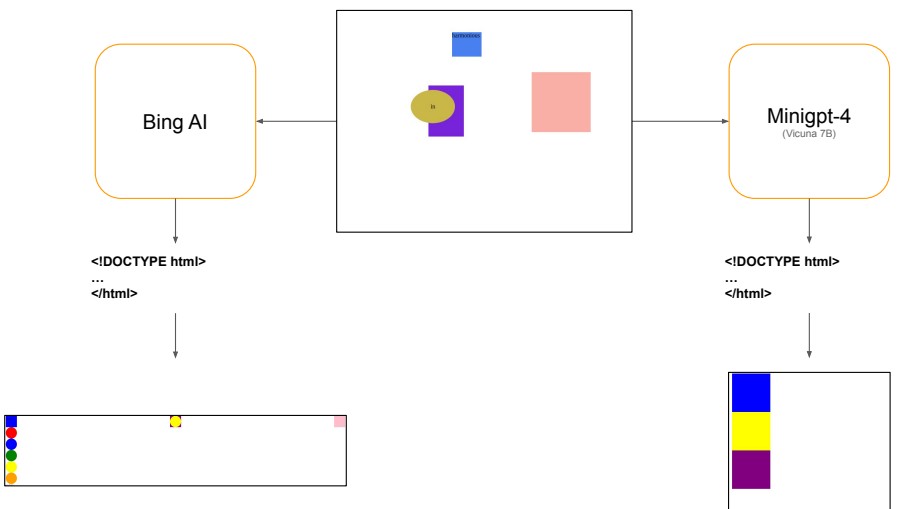

Figure 7: Comparison of minigpt-4 and gpt-4 Bing for mimicking original website design.

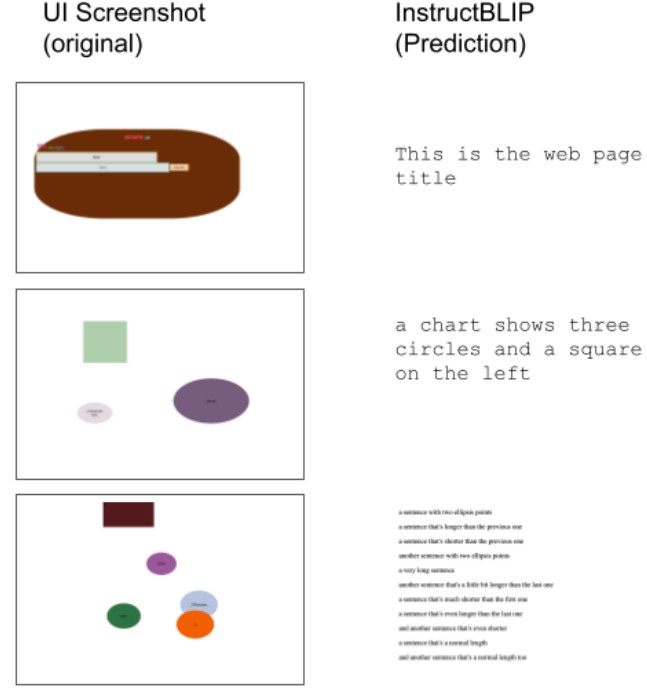

Figure 8: IndstructBlip samples.

```
5       style="margin: 0; position: absolute; top: 50%; left: 50%;
6       transform: translate(-50%, -50%);">been</p>
7   </div>
```

RUID and RUID-Large Datasets

RUID

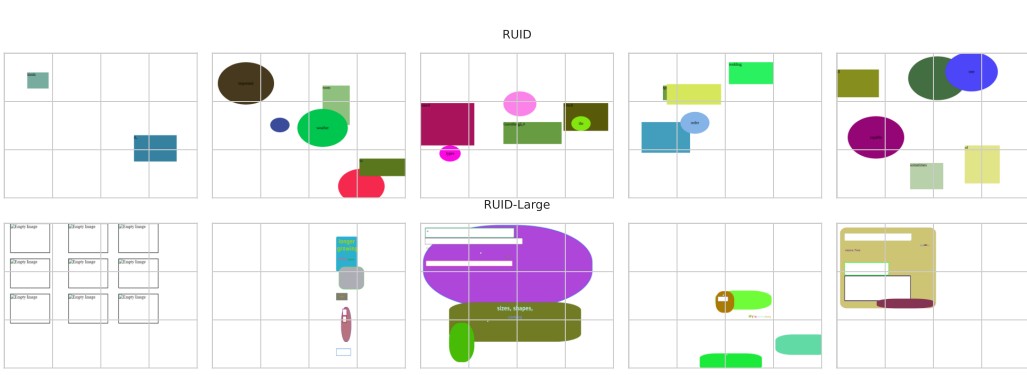

RUID-Large

Figure 9: Some samples from RUID (first row) and RUID-Large (second row) datasets. As we see by this example, RUID is primarily consists of shapes, thus the challenge to the model is accurately predicting the shape and positioning, as well as relations between shapes. In the second row we have RUID-Large which contains a greater diversity of elements to choose form, including images and forms.

### A.3 HUMAN EVALUATION

The survey to evaluate model outputs was filled by 59 volunteers in total, mainly of the student-aged demographic. Each participant was presented with two images at a time, one original image, and one generated image. The participants were not aware which model had generated the image and were asked to rate structural and color similarity using a slider with values in the 0-100 range. Each participant rated 15 images for each dataset, sampled from 100 randomly selected pairs for each model and each dataset.

As a filter, we asked to rate blank images with the highest scores and removed any submissions that did not follow these instructions. This resulted in 3 people's answers being excluded from the analysis. The filter image scores are also removed from the calculations. For other scores, we normalized them for each person using minimum and maximum values and then averaged over the models resulting in the values reported in tables 2 and 4.

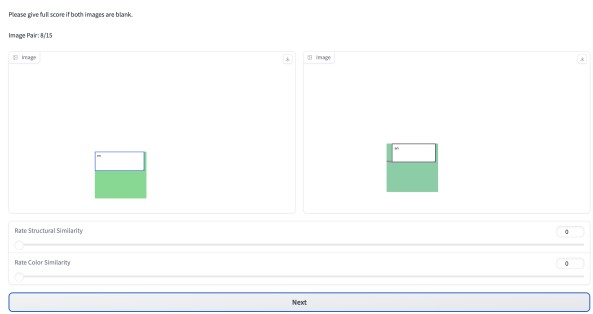

Figure 10: Human evaluation similarity interface.