# OpenReview forum: "Learning UI-to-Code Reverse Generator Using Visual Critic Without Rendering"
_ICLR.cc/2024/Conference — ICLR 2024 Conference Withdrawn Submission_

### Official Review · Reviewer_f5JC · 2023-10-19

**Soundness:** 1 poor
**Presentation:** 2 fair
**Contribution:** 1 poor
**Rating:** 3
**Confidence:** 4

**Summary:**

This paper proposes ViCT, a multimodal Transformer model to reverse engineer UI code from screenshots. An actor-critic framework is used to train the model, addressing the problem of non-differentiable rendering. The model shows superior results on two novel synthetic UI-to-Code datasets, RUID and RUID-Large. A novel metric, htmlBLUE, is proposed to better compare html code.

**Strengths:**

1. The paper works on an interesting problem of automated reverse engineering of HTML/CSS code from UI screenshots.
2. The authors propose to formulate the task as a Reinforcement Learning problem to tackle the problem of non-differentiable web rendering.

**Weaknesses:**

- The paper is not clearly written. Some sections are hard to follow (e.g. Section 3.3).
- Some parts of the paper are inconsistent.
  - In Section 4.1, the authors claim to test InstructBLIP[1] as a baseline, but I could not find it in the experimental results.
  - In Section 4.1, the authors mention an experiment "identifying the number of distinct shapes", which is absent in the paper.
- The main and only datasets the authors use for evaluation are fully synthesized. The UIs in the dataset only contain three types of elements, Rectangle, Ellipse and Button. From the examples in Figure 3, I find them quite unrealistic and do not resemble real-world web UIs, which shadows the effectiveness and practical applicability of the model in genuine scenarios.
- Important details on dataset construction and algorithm design are missing (see Questions).
- Experiments are limited.
  - Missing baselines, e.g. Pix2Struct [2].
  - The models are only evaluated on two synthetic datasets. Can you run experiments on other datasets, such as the dataset of pix2code [3]?
  - "DiT-LLaMA" is missing in Figure 3.

(Minor)
- In Section 1,
> In this paper, we take the first step towards reverse-engineering a UI screenshot, i.e., generating an HTML/CSS code that can reproduce the image.

There are prior works on UI-to-Code tasks, such as Pix2Struct[2] and pix2code[3], as you mentioned in Related Works. Do you mean you are the first to directly generate runable UI code without any postprocessing from images?

- Some typos, e.g. a missing period at the end of Section 2.

[1] Wenliang Dai, Junnan Li, Dongxu Li, Anthony Meng Huat Tiong, Junqi Zhao, Weisheng Wang, Boyang Li, Pascale Fung and Steven Hoi. "InstructBLIP: Towards General-purpose Vision-Language Models with Instruction Tuning." arXiv preprint arXiv:2305.06500. 2023.

[2] Kenton Lee, Mandar Joshi, Iulia Raluca Turc, Hexiang Hu, Fangyu Liu, Julian Martin Eisenschlos, Urvashi Khandelwal, Peter Shaw, Ming-Wei Chang and Kristina Toutanova. "Pix2struct: Screenshot parsing as pretraining for visual language understanding." International Conference on Machine Learning. PMLR. 2023.

[3] Tony Beltramelli. "pix2code: Generating code from a graphical user interface screenshot." Proceedings of the ACM SIGCHI Symposium on Engineering Interactive Computing Systems. 2018.

**Questions:**

1. Please provide more information on the construction of the datasets, RUID and RUID-Large. How do you generate the DOM trees? Which CSS styles do you use as attributes?

2. Please explain the design of the critic model. Is it trained on complete prediction-source pairs and used to estimate values on individual tokens? Additionally, it seems that the critic model only takes visual positional information into account, i.e. IoU. How does the model learn the attributes of the HTML elements, e.g. colors?

3. Please further justify the use of htmlBLEU. In your experiments, you compare htmlBLEU to BLEU with the rendered pixel Mean Squared Error as a standard. Does that mean using MSE as a metric of visual similarity is a better choice?

---

> ### Author Response · Authors · 2023-11-17
>
> We greatly appreciate your detailed review and valuable feedback on our paper and areas of improvement. We will address concerns and questions you raised below.
>
>
> ## Weaknesses
>
> >- The paper is not clearly written...
>
> We will update Section 3.3 in the final draft to expand on the steps of the method.
> > - In Section 4.1, the authors claim to test InstructBLIP[1] ...
>
> While we tested InstructBLIP for multiple samples, the performance was too low for any maningful quantitative metrics, similar to MiniGPT-4. We will provide more examples of the generations.
>
> More specifically, it outputted results such as
>
> ```
> A piece of paper
> ```
> and
> ```
> This is the web page title
> ```
>
> as the source code for an example for 2 squres and a circle and 3 squares respectively. See new Fig. 8.
>
> >- In Section 4.1, the authors mention an experiment ...
>
> The experimental results are reported Table 3 amd Table 4 in terms of the Element N metric. It compares the number of unique elements in the original and resulting code (treating HTML tags as unique elements). We will make it clearer that they are the same metric.
>
> >- The main and only datasets the authors use ...
>
> We use two synthetic datasets RUID and RUID-Large. While RUID contains only the three distinct shapes, **RUID-Large contains almost all possible HTML tags and generates complex DOM trees** and much more closely resembles real world scenarios (for data diversity, they are not always fancy and sometimes need Zoom-in). Figure 3 contains 3 RUID samples and 1 RUID large sample, the updated paper will reflect more samples of both. We will add more RUID large samples.
>
> > - Experiments are limited ...
>
> We do not directly compare to the Pix2Struct since they aim to generate simplified HTML markup. Instead, our goal is to directly generate HTML and CSS code. So the comparison schemes of the two tasks are different. Moreover, their C4 dataset only published URLs but many of them are broken so reproducing their results or comparison to their method is almost impossible. We hope to help alleviate this issue for future research in the field by offering dataset that while synthetic is complex, challennging, fully sharable and reproducible.
>
> Pix2Code aims to generate domain specific language (DSL) and the dataset is curated for that purpose. It only contains elements that can be generated using the intermediate DSL language and thus affords little variance. We generate HTML and CSS code directly and don't have such constraints.
>
> Figure 3 compares models on both RUID and RUID-Large datasets. We will add samples of RUID-Large and add DiT-LLaMA results as well.
>
> ## Questions
> > - Please provide more information on the construction of the datasets ...
>
> The code is generated recusrively by building a tree of maximum depth 6: when expanding each state/node, we draw a random number from (1, 2, 3, 4, 5, 6) as the number of children. Each child is chosen arbitrarily from almost all HTML tags, including a, img, div, span,
> button, text, textarea, input, submit, radio, select, p, checkbox, header. And most common attributes are used, including class, id, style, href, src, alt, title, type, value, placeholder, name, disabled, checked, readonly, selected, multiple, required, as well as css styles such as
> color, background-color, background-image, font-family, font-size, text-align, line-height, text-decoration, margin, padding, border, width, height, position, display, flex etc.
>
> The generator also follows rules to make sure invalid code is not generated. These include avoiding header nesting, such as different header tags being inside each other, oredered and unordered list element placement in ul and ol tags, block vs inline elements: Block-level elements should not be placed inside inline elements, select element usage: ensures proper usage of the select element, textarea and form related rules in that certain elements are not allowed inside forms etc.
>
> > - Please explain the design ...
>
>
> Yes, it predicts IoU similarity class from prediction-source code pairs. Colors are learned through minimizing the CE (cross-entropy) loss instead of the IoU loss, which does not explicitly provide feedback on colors. They work together and produce promising UI-to-code results. We initially tried to train a critic of pixel-level MSE but found training the classification-based critic using IoU scores is more stable.
>
> > - Please further justify the use of htmlBLEU...
>
> When avaliable and affordable, visual metrics tend to be more accurate than code metrics in terms of representing visual similarity. However, rendering webpages from codes requires additional processing (usually a browser instance), resulting extra computation and high latency. While HTMLBleu approximates visual similarity better than BLEU, it is rendering-free and thus much more computationally efficient than visual metrics that requires expensive rendering.

---

### Official Review · Reviewer_TJDi · 2023-10-19

**Soundness:** 3 good
**Presentation:** 3 good
**Contribution:** 3 good
**Rating:** 6
**Confidence:** 2

**Summary:**

The authors propose a novel methodology for retrieving HTML/CSS code from screenshots, by stacking a vision encoder, that parses images as sequence of tokens, to a language decoder, that produce the code itself.
To train their netowork, the authors generate a dataset of simple HTML pages with elements and styles, thus enabling large-scale data generation.
To fine-tune the model, the authors rely on a RL algorithm that tries to maximise the similarity between the renders, formalized as a four-class approach. This formulation is differentiable, thus the method can be fine-tuned with gradient descent.
Also, the authors propose the htmlBLEU metric that emphasizes relevant common pieces of HTML/CSS.
Results show that current state of the art creates allucinations, unable to produce similar results to the ground truth.
The authors clarify that this is a proof of concept, and more must be done to get higher-quality results.

**Strengths:**

1. The paper is original, as it presents an interesrting problem that can be solved through transformers.
2. The state of the art is not able to re-create the same results as the proposal.
3. Interesting technique for generating HTML synthetic data.

**Weaknesses:**

**Why RL?** I understand that it is not possible, given the render, to propagate gradients to the tokens. However, for the same reason, it is not clear from the paper why this is not a problem when optimising the RL policy. The authors should better explain the passage in 3.3, as now it is very confusing to understand.

**Missing ablation study.** The RL algorithm is given some fixed rewards. How the results changes by varying them? And how these values have been chosen?

**Confusion around the htmlBLEU** While the authors write a generic description of the metric, it would be easier for readers to read an algorithm. Also, the proposed metric does not score too different results with respect to BLEU.

**Synthetic data might be harder to parse than real webpages.** While the introduction of the RUID dataset (and its creation) are very interesting and useful, I argue if the randomness of the approach could generate many samples that are very hard to transform to code, thus impeding the improvement of performance at training time.

**Questions:**

1. Can the author better explain why they use RL?
2. Can the author provide a better explanation of the htmlBLEU metric?

---

> ### Author Response · Authors · 2023-11-17
>
> Thank you for your insightful feedback and questions regarding our paper. We appreciate your comments on the originality and potential of our approach and will address the concerns and questions you raised.
>
>
> ## Weaknesses
> > - Why RL? ...
>
> This is not a problem of RL because RL does not require the rendering to be differentiable. It can produce token-level labels by the Q function (i.e., credit assignment), which provide finegrained supervised information.
>
> > - Missing ablation study. The RL algorithm is given some fixed rewards...
>
> These fixed reward values are inspired by CodeRL [1]. While further tuning of these values might further improve the currently achieved performance, our current configuation already show significant advantages over baselines on the task. We will try to tune them for getting a better performance.
>
> [1] Hung Le, Yue Wang, Akhilesh Deepak Gotmare, Silvio Savarese, and Steven CH Hoi. Coderl: Mastering code generation through pretrained models and deep reinforcement learning. arXiv preprint arXiv:2207.01780, 2022
>
> > - Confusion around the htmlBLEU ...
>
> Thank you for the feedback regarding HtmlBLEU. We expand on the defition, and will offer details in the final paper as well as the specifics of the implementation through the code release.
>
> In terms of similarity to BLEU, this is because around 50% of HtmlBLEU scores are composed of BLEU due to alpha choice. This choice is relatively arbitrary anc can be adjusted for specific tasks. Compared to them, HtmlBLEU shows much higher rank correlation to the visual metrics as mentioned in Section 3.4.
>
> **HTMLBLEU** is a composite metric for evaluating HTML document similarity, consisting of the following components:
>
> 1. **Weighted Tag Matching:**
>
>     ```
>     Weighted Tag Match = (Σ_tag min(ref_count, hyp_count) × tag_weight) / (Σ_tag ref_count × tag_weight)
>     ```
>
> 2. **Syntactic DOM Tree Matching:**
>
>     ```
>     DOM Tree Match = Max Weighted Bipartite Matching(DOM Trees)
>     ```
>
> 3. **Attribute Similarity:**
>
>     ```
>     Attribute Similarity = coinciding attributes with matching values / Total Attributes
>     ```
>
> 4. **BLEU Score:**
>
>     ```
>     BLEU Score = sentence_bleu(Reference, Hypothesis)
>     ```
>
> The final **HtmlBLEU** score is calculated as weighted sum of the above parts. The weighted score and attribute similarity allow mitigating score loss due to differences in the structure of the code that has no impact on the visual rendering.
>
>
> > - Synthetic data might be harder to parse than real webpages...
>
> The synthetic data were designed to be diverse so they might contain some unnatural, less fancy, or less organized webpage structures. However, we show that the models are able to learn on the data to a good extent. Compared to scraped content, they provide a way to adjust to the model's inherent token length limit without completely destroying or cutting off part of the webpage structure.
>
>
> ## Questions
> > - Can the author better explain why they use RL?
>
> We appreciate the reviewer raising this important point. Our key motivation for incorporating critic (ViCR) into our training objective is to maximize the visual similarity between the original UI screenshot and the one rendered from the generated code. Specifically, the visual critic allows us to directly optimize the Intersection over Union (IoU) similarity metric through a reward signal, without needing to go through the costly rendering process within every training step (high latency).
>
> As discussed in Section 3.3, the non-differentiable nature of the rendering computation poses challenges for end-to-end optimization of visual fidelity. By training the visual critic to predict the expected IoU similarity from the generated code, we obtain a token-level signal $\hat{q}_\phi\left(w_t^s\right)$ that can guide the encoder-decoder model training.
>
> Note the critic (ViCR) provides a code-level reward $r(W^s)$, while $\hat{q}_\phi\left(w_t^s\right)$ produces a token-level value. **They are different.** It is difficult and also unnecessary to train a token-level reward because standard RL framework is designed to train a Q function $\hat{q}_\phi\left(w_t^s\right)$ to do the credit assignment, i.e., assigning partial credits of the code-level reward  $r(W^s)$ to each token in the code.
>
> Prior works such as CodeRL [1] have shown the benefits of critic-based training for code generation tasks. We adapt the framework for our novel problem setup and dataset.
>
> Compared to our baseline finetuning approach, the proposed training leads to significant gains in visual similarity metrics as shown in Section 4. Overall, we demonstrate the efficacy of visual ciritc for this challenging vision-to-code generation problem.
>
>
> [1] Hung Le, Yue Wang, Akhilesh Deepak Gotmare, Silvio Savarese, and Steven CH Hoi. Coderl: Mastering code generation through pretrained models and deep reinforcement learning. arXiv preprint arXiv:2207.01780, 2022

---

### Official Review · Reviewer_N1mE · 2023-10-30

**Soundness:** 3 good
**Presentation:** 3 good
**Contribution:** 2 fair
**Rating:** 5
**Confidence:** 3

**Summary:**

In this paper, the authors propose a framework to process the screenshots of UI and generate related codes based on the LLM decoder. To solve the problem of inefficiency of rendering, a visual critic without rendering (ViCR) module is introduced to predict visual discrepancy of original and generated UI codes. Also, the paper created two synthetic datasets for training and evaluating. An additional metric, named htmlBLEU score, has been developed to evaluate the UI-to-code performance. The proposed method outperforms previous baseline.

**Strengths:**

1. The paper is well-written and easy to follow.
2. The experimental results are good, demonstrating the effectiveness of proposed method.

**Weaknesses:**

The method is incremental in terms of scientific research value, just simply modifying the normal pattern of inserting vision encoder into language models. The proposed framework is effective in tackling the UI-to-code generation, but not such a fundamental research in representation learning from my perspective.

**Questions:**

The paper claims ViCR has no rendering during fine-tuning, but the training objective is based on IoU between reverse-engineered images and the original UI screenshot. So how to acquire the reverse-engineered images without rendering?

---

> ### Author Response · Authors · 2023-11-17
>
> Thank you for your detailed and insightful review. We appreciate the points you've raised and would like to address some of the concerns mentioned.
>
> ## Weaknesses
> > - The method is incremental in terms of scientific research value, just simply modifying the normal pattern of inserting vision encoder into language models. The proposed framework is effective in tackling the UI-to-code generation, but not such a fundamental research in representation learning from my perspective.
>
> Our main novelty is not on the choices of models but the task, the datasets, and the training framework. UI-to-code reverse engineering is a challenging task that has been rarely explored before. The representations produced by the image encoder and other modules are essential to the success of the task. One primary novelty of our work is the ViCR-based loss and training that produces better representations for the task. It overcomes the weakness of using language loss as a proxy for front-end code. Our other contributions include the novel evaluation metric and two new datasets for the task. Our framework is general: it is not limited to specific choices of vision encoders or language models. So we chose relatively intuitive models to test the general framework.
>
> ## Questions
> > - The paper claims ViCR has no rendering during fine-tuning, but the training objective is based on IoU between reverse-engineered images and the original UI screenshot. So how to acquire the reverse-engineered images without rendering?
>
> We mainly focus on the rendering during the training process when computing a loss for the visual presentation similarity because it is the main bottleneck for training latency and can significantly slowdown the training. We do need offline rendering because they do not affect the training efficiency. The critic model is trained on the rendered samples and used to approximate the IoU similarity class subsequently.

---

### Official Review · Reviewer_o7Uy · 2023-11-04

**Soundness:** 2 fair
**Presentation:** 3 good
**Contribution:** 2 fair
**Rating:** 5
**Confidence:** 4

**Summary:**

This paper provides ViCT (Vision-Code Transformer), an UI-conditioned code generation model that is fine-tuned with reinforcement learning (RL). More specifically, ViCT takes an UI image as input and generates HTML. ViCT consists of vision foundations models (e.g., ViT and DiT) for encoding images and Large Language Models (LLMs) for generating code. To further align ViCT with the visual similarity between an input UI image and an UI image rendered by generated code, this paper provides ViCR (Visual Critic without Rendering), a reward model for RL fine-tuning. To demonstrate the proof of concept, this paper builds RUID (Random UI Dataset), a new dataset for UI to code generation, that includes about 50K pair of UI image and HTML. With the dataset, this paper shows that ViCT provides comparable performance and fine-tuning with ViCR can further improves the performance.

**Strengths:**

- S1. The main idea of fine-tuning an image-conditioned text generation model with a reward model and reinforcement learning is very interesting. Even though the concept of an image-conditioned code generation was proposed before, using foundation models (DiT and Llama) and fine-tuning the model with RL (Policy Gradient method) seems novel.

- S2. To demonstrate the proof of concept, this paper builds a new dataset for UI to code generation, which contains about 50K pairs of UI and HTML (RUID-Large, Random UI Dataset).

**Weaknesses:**

- W1. Overall architecture of the proposed method (ViCT) seems reasonable. However, I am not sure that the design choice for the reward modeling and RL fine-tuning is effective. The overall method is similar to Reinforcement Learning with Human Feedback (RLHF), a recent prevailing method for LLM alignment. In RLHF, the reward model (RM) is usually modeled by relative feedback (preference or superiority) over a pair of inputs. Also, the prevalent RL algorithm is Proximal Policy Optimization (PPO) rather than vanilla Policy Gradient (PG). It would be better to provide some considerations on these design choices. And, it would be much better to provide a comparison between ViCR (absolute feedback + PG) and RLHF methods (relative feedback + PPO).

- W2. I am not sure how effectively ViCR models an intermediate reward in Eq 2. According to Eq 2., \hat{q_theta}(w_t^s), a value function for the token w_t^s is used. Can the reward model (ViCR) estimate the value for an intermediate token in partially generated code?

**Questions:**

- Q1. Regarding W1, how does the reward model (ViCR) perform? Since this paper models ViCR as classification of visual similarity (very low, low, high and very high), it will be better to provide classification accuracy.

- Q2. Regarding W1, how does the learning curve (e.g., IoU over learning steps) look like? It will help readers to understand the learning dynamics in RL fine-tuning of ViCT.

- Q3. Regarding W2, how does the token-level reward model perform?

---

> ### Author Response · Authors · 2023-11-17
>
> Thank you for your constructive feedback and insightful questions regarding our paper on the Vision-Code Transformer (ViCT). We appreciate the opportunity to clarify and elaborate on the aspects you've highlighted.
>
>
> ## Weaknesses
> > - W1. Overall architecture of the proposed method (ViCT) seems reasonable. However, I am not sure that the design choice for the reward modeling and RL fine-tuning is effective. The overall method is similar to Reinforcement Learning with Human Feedback (RLHF), a recent prevailing method for LLM alignment. In RLHF, the reward model (RM) is usually modeled by relative feedback (preference or superiority) over a pair of inputs. Also, the prevalent RL algorithm is Proximal Policy Optimization (PPO) rather than vanilla Policy Gradient (PG). It would be better to provide some considerations on these design choices. And, it would be much better to provide a comparison between ViCR (absolute feedback + PG) and RLHF methods (relative feedback + PPO).
>
> We appreciate the reviewers feedback and the insight of RLHF. RLHF is a really interesting next step for the research. However, it does not learn from human feedback; it works in the vision-language domain (RLHF focuses on language modeling); and it addresses a problem (UI-to-code reverse engineering) different from that of RLHF (human-LLM alignment).
>
> In terms of efficiency, RLHF is a lot more expensive, specially for collecting human preference annotations. Our method offers a low-cost alternative that gives significant improvements in performance. In terms of the critic model, our method does not need a fine-grained or accurate critic as RLHF. This is important because there does not exist an accurate critic model (neither close-source nor open-source models) for the UI-to-code task, as shown in the examples of MiniGPT4, Bing in Appendix A1, Fig 7.
>
> > - W2. I am not sure how effectively ViCR models an intermediate reward in Eq 2. According to Eq 2., $\hat{q_\theta}(w_t^s)$, a value function for the token w_t^s is used. Can the reward model (ViCR) estimate the value for an intermediate token in partially generated code?
>
> Reward model and Q function are different: the former can assigns only one reward to each trajectory while the latter learns to assign a value (credit-assignment) to every token. ViCR is our reward model that produces one label for each generated code. Our Q function $\hat{q_\theta}(w_t^s)$ is a finetuned GPT2 producing token-wise value.
>
> ## Questions
> > - Q1. Regarding W1, how does the reward model (ViCR) perform? Since this paper models ViCR as classification of visual similarity (very low, low, high and very high), it will be better to provide classification accuracy.
>
> The classification accuracy of ViCR is 0.78. As shown by the final UI-to-code results, the accuracy is high enough for train the ViCT model. Notably most errors are in neighboaring classes as seen in the Figure 4, For example the model might mistake very low, and low cases, however, more drastic mistakes between low and very high classes are very rare.
>
> > - Q2. Regarding W1, how does the learning curve (e.g., IoU over learning steps) look like? It will help readers to understand the learning dynamics in RL fine-tuning of ViCT.
>
> We appreciate the reviewers suggestion of looking at the metric. Unfortunately, we did not track this metric during initial training, since it would signficiantly slow down the training process. We do not render the images that would be required for IoU calculation between training steps. but we have added it to the pipeline and will provide it for future models.
>
> > - Q3. Regarding W2, how does the token-level reward model perform?
>
> It would be very interesting to explore the token-level reward models. One such could be sequence level reward, which would modify weights of each given sample improving the baseline performance.